# Mixture-of-Agents Enhances Large Language Model Capabilities

**Junlin Wang**[1]*  **Jue Wang**[2]  **Ben Athiwaratkun**[2]  **Ce Zhang**[2,3]  **James Zou**[2,4]

[1]Duke University  [2]Together AI  [3]University of Chicago  [4]Stanford University
junlin.wang2@duke.edu, {jue, ben}@together.ai
cez@uchicago.edu, jamesz@stanford.edu

## Abstract

Recent advances in large language models (LLMs) demonstrate substantial capabilities in natural language understanding and generation tasks. With the growing number of LLMs, how to harness the collective expertise of multiple LLMs is an exciting open direction. Toward this goal, we propose a new approach that leverages the collective strengths of multiple LLMs through a Mixture-of-Agents (MoA) methodology. In our approach, we construct a layered MoA architecture wherein each layer comprises multiple LLM agents. Each agent takes all the outputs from agents in the previous layer as auxiliary information in generating its response. MoA models achieves state-of-art performance on AlpacaEval 2.0, Arena-Hard, MT-Bench, and FLASK, surpassing GPT-4 Omni. For example, our MoA using only open-source LLMs achieves a score of 65.1% on AlpacaEval 2.0 compared to 57.5% by GPT-4 Omni.[1]

## 1 Introduction

Large language models (LLMs) (Zhang et al., 2022a; Chowdhery et al., 2022; Touvron et al., 2023a; Team et al., 2023; Brown et al., 2020; OpenAI, 2023) have significantly advanced the field of natural language understanding and generation in recent years. These models are pretrained on vast amounts of data and subsequently aligned with human preferences to generate helpful and coherent outputs (Ouyang et al., 2022). However, despite the plethora of LLMs and their impressive achievements, they still face inherent constraints on model size and training data. Further scaling up these models is exceptionally costly, often requiring extensive retraining on several trillion tokens.

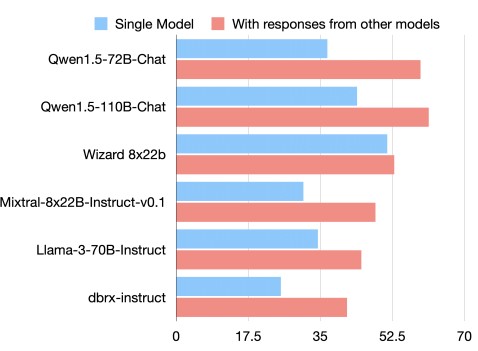

Figure 1: AlpacaEval 2.0 LC win rates improve when provided with responses from the six models in this figure. Table 1 presents the template.

At the same time, different LLMs possess unique strengths and specialize in various tasks aspects. For instance, some models excel at complex instruction following (Xu et al., 2023a) while others may be better suited for code generation (Roziere et al., 2023; Guo et al., 2024). This diversity in skill sets among different LLMs presents an intriguing question: *Can we harness the collective expertise of multiple LLMs to create a more capable and robust model?* Our answer to this question is *Yes*. We identify an inherent phenomenon we term the *collaborativeness* of LLMs — wherein an LLM tends to generate better responses when presented with outputs from other models, even if these other models are less capable by itself. Figure 1 showcases the LC win rate on the AlpacaEval 2.0 benchmark (Dubois et al., 2024) for 6 popular LLMs. We first tested each model independently, and then use each of them as an aggregator to

---

*Work done while interning at Together AI

[1]https://github.com/togethercomputer/moa.

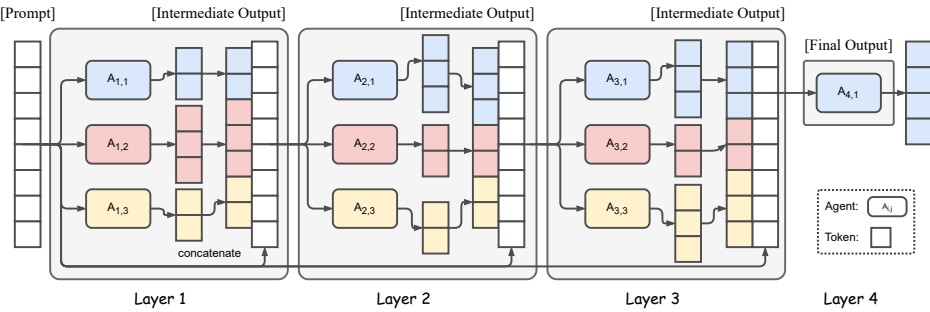

Figure 2: Illustration of the Mixture-of-Agents Structure. This example shows 4 MoA layers where the first layer has 3 proposers, the second and third layer have 3 aggregators that also serve as proposers for the next layer, and the last layer has one aggregator.

combine their outputs. We found when a model is provided with answers generated by other models, its LC win rate significantly improves. This indicates that the collaborativeness phenomenon is widespread among LLMs. Remarkably, this improvement occurs even when the auxiliary responses provided by the other models are of lower quality than what an individual LLM could generate independently.

Based on this finding, this paper introduces a Mixture-of-Agents (MoA) methodology that leverages multiple LLMs to iteratively enhance the generation quality. The structure of MoA is illustrated in Figure 2. Initially, LLMs in the first layer, denoted as Agents $A_{1,1}, ...A_{1,n}$ independently generate responses to a given prompt. These responses are then presented to agents in the next layer $A_{2,1}, ...A_{2,n}$ (which may reuse a model from the first layer) for further refinement. This iterative refinement process continues for several cycles until obtaining a more robust and comprehensive response.

To ensure effective collaboration among models and improve overall response quality, careful selection of LLMs for each MoA layer is crucial. This selection process is guided by two primary criteria: (a) Performance Metrics: The win rate of each model plays an important role in determining their inclusion in MoA. (b) Diversity Considerations: The diversity of model outputs is also crucial, and responses generated by heterogeneous models contribute significantly more than those produced by the same model, as we show later in section 3.3. By leveraging these criteria — performance and diversity — MoA aims to mitigate individual model deficiencies and enhance overall response quality through collaborative synthesis.

We conduct comprehensive evaluations using AlpacaEval 2.0, Arena-Hard (Li et al., 2024), MT-Bench (Zheng et al., 2023), FLASK (Ye et al., 2023) benchmarks for assessing the response quality across various dimensions. The results demonstrate substantial improvements with our proposed method, achieving SOTA win rate of 65.8% on AlpacaEval 2.0, outperforming GPT-4 Omni.

The **contributions** of this work are summarized as follows: (1) *Novel framework*: we propose a Mixture-of-Agents framework designed to leverage the strengths of multiple LLMs, thereby improving their reasoning and language generation capabilities. (2) *Finding of collaborativeness of language models*: we highlight the inherit collaborativeness among LLMs, where models tend to generate better quality responses when they have access to outputs from other models, even if those outputs are of lower quality. (3) *State-of-the-art LLM performance*: we conducted extensive experiments using multiple highly-competitive benchmarks such as AlpacaEval 2.0, MT-Bench, and FLASK; our MoA framework achieves state-of-the-art performance on these benchmarks.

## 2 MIXTURE-OF-AGENTS METHODOLOGY

In this section, we present our proposed methodology for leveraging multiple models to achieve boosted performance. We begin by demonstrating that LLMs possess collaborativeness and thus can improve their responses based on the outputs of other models. Following this, we introduce the Mixture-of-Agents methodology and discuss its design implications.

## 2.1 COLLABORATIVENESS OF LLMS

We begin by demonstrating the collaborativeness of LLMs, specifically their ability to generate higher quality responses when they can reference outputs from other models. As we have shown in the introduction and Figure 1, many of today's available LLMs exhibit this collaborative capability.

An important pathway to extract maximum benefits from collaboration of multiple LLMs is to characterize how different models are good at in various aspects of collaboration. During the collaboration process, we can categorize LLMs into two distinct roles:

**Proposers** excel at generating useful reference responses for use by other models. While a good proposer may not necessarily produce responses with high scores by itself, it should offer more context and diverse perspectives, ultimately contributing to better final responses when used by an aggregator.

**Aggregators** are models proficient in synthesizing responses from other models into a single, high-quality output. An effective aggregator should maintain or enhance output quality even when integrating inputs that are of lesser quality than its own.

Section 3.3 empirically validate the roles of aggregators and proposers. Specifically, we show that many LLMs possess capabilities both as aggregators and proposers, while certain models displayed specialized proficiencies in distinct roles. GPT-4o, Qwen1.5, LLaMA-3 emerged as a versatile model effective in both assisting and aggregating tasks. In contrast, WizardLM demonstrated excellent performance as an proposer model but struggled to maintain its effectiveness in aggregating responses from other models.

Given that an aggregator can generate higher-quality responses by building upon outputs from other models, we propose further enhancing this collaborative potential by introducing additional aggregators. One intuitive idea is to replicate the exercise with multiple aggregators — initially using several to aggregate better answers and then re-aggregating these aggregated answers. By incorporating more aggregators into the process, we can iteratively synthesize and refine the responses, leveraging the strengths of multiple models to produce superior outcomes. This leads to the design of our proposed Mixture-of-Agents.

## 2.2 MIXTURE-OF-AGENTS

The structure of MoA is illustrated in Figure 2. It has $l$ layers and each layer-$i$ consists of $n$ LLMs, denoted by $A_{i,1}$, $A_{i,2}$, ..., $A_{i,n}$. It is important to note that LLMs can be reused either within the same layer or across different layers. When many LLMs in a layer are identical, this configuration leads to a special structure that corresponds to a model generating multiple possibly different outputs (due to the stochasticity of temperature sampling). We refer to this setting as single-proposer, where only a sparse subset of models are activated.

Here, each LLM $A_{i,j}$ processes an input text and generates its continuation. Our method does not require any fine-tuning and only utilizes the interface of prompting and generation of LLMs. Formally, given an input prompt $x_1$, the output of $i$-th MoA layer $y_i$ can be expressed as follows:

$$y_i = \oplus_{j=1}^n [A_{i,j}(x_i)] \ + x_1, x_{i+1} = y_i \tag{1}$$

where $+$ here means concatenation of texts; $\oplus$ means application of the Aggregate-and-Synthesize prompt shown in Table 1 to these model outputs. Table 1 represents the template of system prompt, and the original user query is included immediately after this system prompt.

In practice, we do not need to concatenate prompt and all model responses so only one LLM is needed to be used in the last layer. Therefore, we use the output of an LLM from the $l$-th layer ($A_{l,1}(x_l)$) as the final output and evaluate the metrics based on it.

## 2.3 ANALOGY TO MIXTURE-OF-EXPERTS

Mixture-of-Experts (MoE) (Shazeer et al., 2017) is a prominent and well-established technique in machine learning where multiple expert networks specialize in different skill sets. The MoE approach has shown significant success across various applications due to its ability to leverage

Table 1: Aggregate-and-Synthesize Prompt to integrate responses from other models.

---

You have been provided with a set of responses from various open-source models to the latest user query. Your task is to synthesize these responses into a single, high-quality response. It is crucial to critically evaluate the information provided in these responses, recognizing that some of it may be biased or incorrect. Your response should not simply replicate the given answers but should offer a refined, accurate, and comprehensive reply to the instruction. Ensure your response is well-structured, coherent, and adheres to the highest standards of accuracy and reliability.

Responses from models:
1. [Model Response from $A_{i,1}$]
2. [Model Response from $A_{i,2}$]
...
$n$. [Model Response from $A_{i,n}$]

---

diverse model capabilities for complex problem-solving tasks. Our MoA method draws inspiration from this methodology.

A typical MoE design consists of a stack of layers known as MoE layers. Each layer comprises a set of $n$ expert networks alongside a gating network and includes residual connections for improved gradient flow. Formally, for layer $i$, this design can be expressed as follows:

$$y_i = \sum_{j=1}^{n} G_{i,j}(x_i) E_{i,j}(x_i) + x_i \qquad (2)$$

where $G_{i,j}$ represents the output from the gating network corresponding to expert $j$, and $E_{i,j}$ denotes the function computed by expert network $j$. The leverage of multiple experts allows the model to learn different skill sets and focus on various aspects of the task at hand. The gating network $G$ dynamically routes to the appropriate experts, enabling efficient utilization of computational resources by activating only the specialized sub-networks necessary.

From a high-level perspective, our proposed MoA framework extends the MoE concept to the model level by operating at the model level rather than at the activation level. Specifically, our MoA approach leverages LLMs and operates entirely through the prompt interface rather than requiring modifications to internal activations or weights. This means that instead of having specialized sub-networks within a single model like in MoE, we utilize multiple full-fledged LLMs across different layers. Note that in our approach, we consolidate the roles of the gating network and expert networks using a LLM, as the intrinsic capacity of LLMs allows them to effectively regularize inputs by interpreting prompts and generating coherent outputs without needing external mechanisms for coordination. For this work, we design and evaluate MoA in a dense configuration, where all "expert" LLMs in the network process the inputs. However, the method can be seamlessly extended to a sparse configuration, dynamically selecting which LLMs to generate.

Moreover, since this method relies solely on prompting capabilities inherent within off-the-shelf models: (1) It eliminates computational overhead associated with fine-tuning; (2) It provides flexibility and scalability: our method can be applied to the latest LLMs regardless of their size or architecture.

## 3 EVALUATION

This section presents a comprehensive evaluation of our proposed MoA. Our findings show that:

1. We achieve significant improvements on AlpacaEval 2.0, Arena-Hard, MT-Bench, and FLASK benchmarks. Notably, with open-source models only, our approach outperforms GPT-4o on AlpacaEval 2.0, MT-Bench, and FLASK.

2. We conduct extensive experiments to provide better understandings of the internal mechanism of MoA.

3. Through a detailed budget analysis, several implementations of MoA can deliver better performance to GPT-4 Turbo while being $2\times$ more cost-effective.

## 3.1 SETUP

**Benchmarks**  We mainly evaluate models on AlpacaEval 2.0 (Dubois et al., 2024), a leading benchmark for assessing the alignment of LLMs with human preferences. It contains 805 instructions representative of real use cases. Each model's response is directly compared against that of the GPT-4 (`gpt-4-1106-preview`), with a GPT-4-based evaluator determining the likelihood of preferring the evaluated model's response. To ensure fairness, the evaluation employs length-controlled (LC) win rates, effectively neutralizing length bias.[2]

Additionally, we also evaluate on Arena-Hard (Li et al., 2024), MT-Bench (Zheng et al., 2023) and FLASK (Ye et al., 2023). Arena-Hard evaluates performance on 500 challenging user queries, encompassing a diverse range of topics such as coding, mathematics, and logic puzzles. MT-Bench uses GPT-4 to grade and give a score to model's answer. FLASK, on the other hand, offers a more granular evaluation with 12 skill-specific scores.

**Models**  In our study, we constructed our default MoA by using only open-source models to achieve competitive performance. The models included are: Qwen1.5-110B-Chat (Bai et al., 2023), Qwen1.5-72B-Chat, WizardLM-8x22B (Xu et al., 2023a), LLaMA-3-70B-Instruct (Touvron et al., 2023b), Mixtral-8x22B-v0.1 (Jiang et al., 2024), dbrx-instruct (The Mosaic Research Team, 2024). We construct 3 MoA layers and use the same set of models in each MoA layer. We use Qwen1.5-110B-Chat as the aggregator in the last layer. We also developed a variant called MoA w/ GPT-4o, which prioritizes high-quality outputs by using GPT-4o as the aggregator in the final MoA layer. Another variant, MoA-Lite, emphasizes cost-effectiveness. It uses the same set of models as proposers but includes only 2 MoA layers and employs Qwen1.5-72B-Chat as the aggregator. This makes it more cost-effective than GPT-4o while achieving a $1.8\%$ improvement in quality on AlpacaEval 2.0. We ensure strict adherence to the licensing terms of all models utilized in this research. For open-source models, all inferences were ran through Together Inference Endpoint.[3] We mainly use large models to prioritize accuracy here, but we observed similar performance improvements with smaller models as well. The results can be found in Appendix A.

## 3.2 BENCHMARK RESULTS

In this subsection, we present our evaluation results on three standard benchmarks: AlpacaEval 2.0, Arena-Hard, MT-Bench, and FLASK. These benchmarks were chosen to comprehensively assess the performance of our approach and compare with the state-of-the-art LLMs.

**AlpacaEval 2.0**  We conducted comparisons against leading models such as GPT-4 and other state-of-the-art open-source models. The detailed results are presented in Table 2 where our MoA methodology achieved top positions on the AlpacaEval 2.0 leaderboard, demonstrating a remarkable $8.2\%$ absolute improvement over the previous top model, GPT-4o. Moreover, it is particularly noteworthy that our model outperformed GPT-4o using solely open-source models, achieving a margin of $7.6\%$ absolute improvement from $57.5\%$ (GPT-4o) to $65.1\%$ (MoA). Our MoA-Lite setup uses less layers and being more cost-effective. Even with this lighter approach, we still outperform the best model by $1.8\%$, improving from $57.5\%$ (GPT-4o) to $59.3\%$ (MoA-Lite). This further highlights the effectiveness of our method in leveraging open-source models capabilities with varying compute budget to their fullest potential.

**Arena-Hard**  In the Arena-Hard benchmark, our MoA with GPT-4o achieved a $90.3\%$ score, significantly outperforming GPT-4o's single model score $79.2\%$. Even the more cost-effective MoA-Lite variant showed strong results with a $71.3\%$ win rate. These results highlight the robustness and efficiency of our approach in challenging prompts.

**MT-Bench**  Though improvements over individual models on the MT-Bench are relatively incremental, this is understandable given that current models already perform exceptionally well on this benchmark, as a single model alone can achieve scores greater than 9 out of 10. Despite the marginal

---

[2]This metric tracks closely with human preferences, achieving a Spearman correlation of 0.98 with actual human evaluations (Dubois et al., 2024).

[3]https://api.together.ai/playground/chat

Table 2: Results on AlpacaEval 2.0, Arena-Hard, and MT-Bench. MoA and MoA-Lite correspond to the 6 proposers with 3 layers and with 2 layers respectively. MoA w/ GPT-4o corresponds to using GPT-4o as the final aggregator in MoA. MoA-Lite-Single has the same aggregator but only contains one proposer which is Qwen1.5 110B Chat. We ran our experiments three times and reported the average scores along with the standard deviation. [†] denotes our replication of the AlpacaEval results. We ran all the MT-Bench scores ourselves to get turn-based scores. 'Agg.' denotes the aggregated score with formula: $(AlpacaEval\text{-}LC\text{-}win + Arena\text{-}Hard\text{-}win + MT\text{-}Bench\text{-}score \times 10)/3$.

| Model | Agg. | AlpacaEval 2.0 | | Arena-Hard | MT-Bench | | |
| --- | --- | --- | --- | --- | --- | --- | --- |
| | | LC win. | win. | win. | Avg. | 1st turn | 2nd turn |
| MoA w/ GPT-4o | 83.3 | $65.7_{\pm0.7}\%$ | $78.7_{\pm0.2}\%$ | $90.3_{\pm0.5}\%$ | $9.40_{\pm0.06}$ | 9.49 | 9.31 |
| MoA | 78.3 | $65.1_{\pm0.6}\%$ | $59.8_{\pm0.3}\%$ | $77.4_{\pm0.5}\%$ | $9.25_{\pm0.10}$ | 9.44 | 9.07 |
| GPT-4 Turbo (04/09) | 76.7 | 55.0% | 46.1% | 82.0 | 9.31 | 9.35 | 9.28 |
| GPT-4 Omni (05/13) | 76.2 | 57.5% | 51.3% | 79.2 | 9.19 | 9.31 | 9.07 |
| MoA-Lite | 74.1 | $59.3_{\pm0.2}\%$ | $57.0_{\pm0.7}\%$ | $71.3_{\pm0.7}\%$ | $9.18_{\pm0.09}$ | 9.38 | 8.99 |
| GPT-4 Preview (11/06) | 73.6 | 50.0% | 50.0% | 78.7 | 9.20 | 9.38 | 9.03 |
| WizardLM 8x22B[†] | 70.1 | 51.3% | 62.3% | 71.3 | 8.78 | 8.96 | 8.61 |
| MoA-Lite-Single | 64.7 | 47.8% | 37.9% | 59.5 | 8.69 | 9.19 | 8.19 |
| Qwen1.5 110B Chat | 63.3 | 43.9% | 33.8% | 56.4 | 8.96 | 9.23 | 8.63 |
| Llama 3 70B Instruct | 56.8 | 34.4% | 33.2% | 46.6 | 8.94 | 9.2 | 8.68 |
| GPT-4 (03/14) | 53.9 | 35.3% | 22.1% | 37.9 | 8.84 | 9.08 | 8.61 |
| Qwen1.5 72B Chat | 52.4 | 36.6% | 26.5% | 36.1 | 8.44 | 8.55 | 8.34 |
| | | | % | 36.4 | 8.78 | 9.11 | 8.44 |

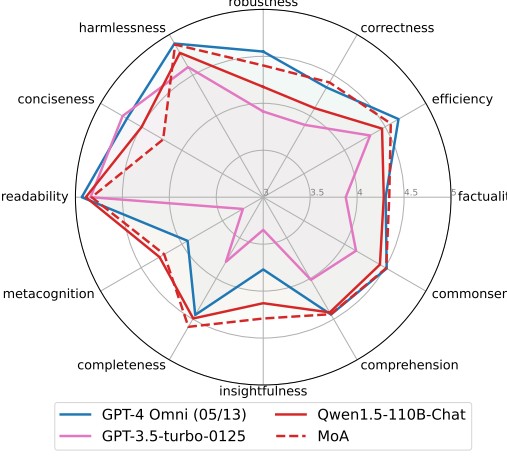

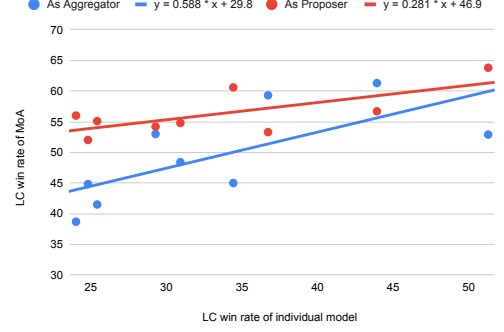

Figure 3: Results on FLASK where we use the default 6-proposer MoA setup and Qwen1.5-110B is the aggregator. We include the results of GPT-3.5, GPT-4o, and Qwen1.5-110B when used as standalone models for comparison.

Figure 4: The 'As Aggregator' curve represents the score when the selected model serves as aggregator, using the 6 proposers from the default MoA setup; The 'As Proposer' curve depicts the score when the selected model serves as proposer, generating proposed answers 6 times, with Qwen1.5-110B as aggregator.

enhancements, our approach still secures the top position on the leaderboard. This demonstrates that even with already highly optimized benchmarks, our method can push the boundaries further, maintaining the leadership.

**FLASK** FLASK provides fine-grained evaluation of models. Among those metrics, MoA excels in several key aspects. Specifically, our methodology shows improvement in robustness, correctness, insightfulness, compared to the single model score of the aggregator, Qwen-110B-Chat. Additionally, MoA also outperforms GPT-4 Omni in terms of insightfulness, correctness, factuality, completeness, and metacognition. One metric where MoA did not do as well was conciseness; the model produced outputs that were moderately more verbose.

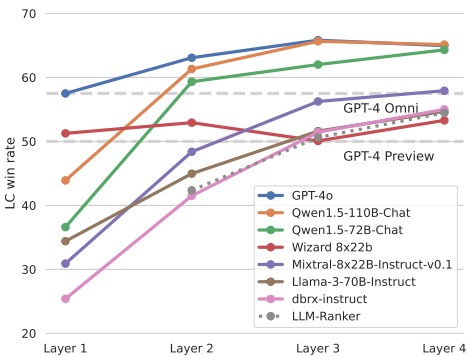 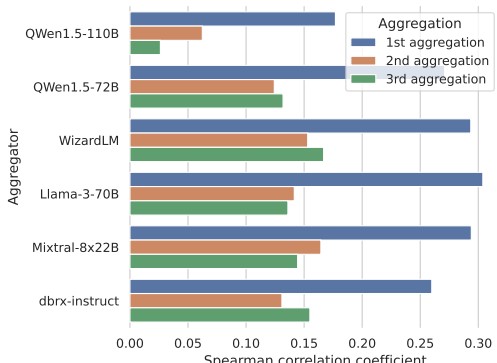

Figure 5: (a) LC win rate on AlpacaEval 2.0 with different aggregators in the 6-model Mixture-of-Agents setup. All the curves use the same 6 proposer agents; they only differ in the choice of the final aggregator. The LLM ranker uses Qwen1.5-110B-Chat model with a prompt format in Appendix Appendix C. The GPT-4o model is only used to aggregate the output for the purpose of evaluation and does not participate as a proposer towards the next layer. (b) Spearman correlation between BLEU scores (calculated using 3-gram, 4-gram, and 5-gram metrics) and win rate of the proposed outputs.

### 3.3    WHAT MAKES MIXTURE-OF-AGENTS WORK WELL?

In this subsection, we conduct experiments that provide us better understandings of the internal mechanism of Mixture-of-Agents. We summarize key insights below.

**Mixture-of-Agents significantly outperforms LLM rankers.**    First, we compare Mixture-of-Agents with an LLM-based ranker which uses the aggregator model to select one of the answers that are generated by the proposers, instead of generating a new output. The results are shown in Figure 5, where we can observe that the MoA approach significantly outperforms an LLM-ranker baseline. The fact that MoA outperforms the ranking approach suggests that the aggregator does not simply select one of the generated answers by the proposers, but potentially performs more sophisticated aggregation over all proposed generations.

**MoA tends to incorporate the best proposed answers.**    We also compare the aggregator's response with the proposers' responses via similarity scores such as BLEU (Papineni et al., 2002) which reflects n-gram overlaps. Within each sample, given $n$ proposed answers by the proposers, we calculate the Spearman's rank correlation coefficient between the $n$ similarity scores and the $n$ preference scores determined by the GPT-4 based evaluator. The results in Figure 5 indeed confirms a positive correlation between the win rate and the BLEU score. We also provide results with Levenshtein similarity (RapidFuzz, 2023) or TF-IDF as opposed to BLEU scores in Appendix B. where both alternative approaches for textual similarities also yield positive correlation with the preference scores. For similarity comparison between proposers and aggregator, can refer to Appendix G.

**Relationship of single model win rates to the MoA system.**    Our results indicate a positive relationship between the individual LLM's performance in each role (proposer or aggregator) and the final performance of MoA. Through linear regression analysis, presented in Figure 4, with the x-axis representing the model's performance as a proposer/aggregator (we adopt adopt the Single-Proposer setting for the proposer one), and the y-axis representing the MoA's final performance, we observed that the regression coefficient for the aggregator model (0.588) is higher than that for the proposer model (0.281). This suggests that high-quality models are useful for both the aggregator and proposer roles. The steeper slope for the aggregator fit suggests that the final MoA performance is more sensitive to the quality of the aggregator model than the proposer.

**Effect of model diversity and the number of proposers.**    We analyze how the number of proposals affect the final output quality by varying $n$, the number of proposers in each layer. We show the results in Table 3 where we find that scores increases monotonically with $n$, reflecting the benefits of having more auxiliary information. In addition, we also quantify the impact of using a diverse

Table 3: Effects of the number of proposer models on AlpacaEval 2.0. We denote $n$ as either the number of models in an MoA layer or the number of proposed outputs in the single-proposer setting. We use Qwen1.5-110B-Chat as the aggregator and use 2 MoA layers for all settings in this table.

| Setting | Multiple-Proposer | Single-Proposer |
|---|---|---|
| $n = 6$ | 61.3% | 56.7% |
| $n = 3$ | 58.0% | 56.1% |
| $n = 2$ | 58.8% | 54.5% |
| $n = 1$ | 47.8% | 47.8% |

Table 4: Impact of different models serving as proposers vs aggregators. When evaluating aggregators, all six models serve as proposers; when evaluating proposers, Qwen1.5-110B-Chat serves as the aggregator. We use 2 MoA layers here.

| Model | As aggregator | As proposer |
|---|---|---|
| Qwen1.5-110B-Chat | 61.3% | 56.7% |
| Qwen1.5-72B-Chat | 59.3% | 53.3% |
| LLaMA-3-70b-Instruct | 45.0% | 60.6% |
| WizardLM 8x22B | 52.9% | 63.8% |
| Mixtral-8x22B-Instruct | 48.4% | 54.8% |
| dbrx-instruct | 41.5% | 55.1% |

Table 5: Comparison of Multi-Agent Methods.

| Method | # Aggregation | AlpacaEval2 (LC) | Avg Cost ($) |
|---|---|---|---|
| Standalone | 0 | 43.9 | - |
| MoA | 1 | 61.3 | 0.00852 |
| | 2 | 65.7 | 0.03150 |
| MAD | 1 | 53.5 | 0.00819 |
| | 2 | 50.6 | 0.03140 |
| Reconcile | 1 | 47.6 | 0.00818 |
| | 2 | 47.7 | 0.02910 |

set of LLMs as proposers. For each $n$, we compare two settings: "single-proposer" where the $n$ responses are generated by the same LLM with a temperature of 0.7; and "multiple-proposer" where each response is generated by a different LLMs. Overall, using multiple different LLMs consistently yielded better results. Both results suggest that having a larger number of diverse LLM agents in each MoA layer can improve performance. Further scaling the width of MoA is a promising direction of future investigation.

**Specialization of models in the MoA ecosystem.** We also conducted experiments to determine which models excel in specific roles. Specifically, Table 4 shows that GPT-4o, Qwen, LLaMA-3 emerged as a versatile model effective in both assisting and aggregating tasks. In contrast, WizardLM demonstrated excellent performance as an proposer model but struggled to maintain its effectiveness in aggregating responses from other models.

**Comparison to Existing Multi-Agent Methods** We conducted comparative experiments to evaluate MoA against existing multi-agent methods like MAD Liang et al. (2023) and Reconcile Chen et al. (2023a), which typically focus on tasks with short, deterministic answers and are not directly applicable to open-ended, contextually rich chat scenarios. We adapted their prompts for chatting and removed features like JSON outputs and confidence levels for Reconcile.

Our results (Table 5) show that MoA consistently outperforms MAD and Reconcile. With one round of aggregation, MoA achieves an AlpacaEval2 score of 61.3, surpassing MAD's 53.5 and Reconcile's 47.6. This gap widens with two rounds, where MoA scores 65.7 compared to MAD's 50.6 and Reconcile's 47.7. MoA scales better with more aggregation rounds, and maintain cost efficiency, achieving better performance within similar budget constraints.

## 3.4 BUDGET AND TOKEN ANALYSIS

To understand the relationship between budget, token usage, and LC win rates, we conducted a budget and token analysis. Figure 6a and Figure 6b illustrate these relationships.

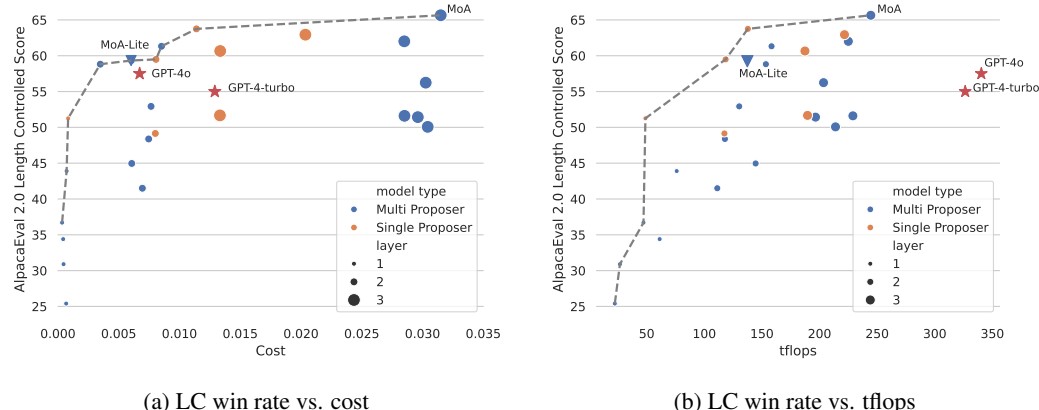

(a) LC win rate vs. cost            (b) LC win rate vs. tflops

Figure 6: (a) Performance trade-off versus cost. The dots (layer > 2) with the same size indicate MoA with different final aggregator. (b) Performance trade-off versus the number of tera floating operations (tflops), which we use as a proxy for latency. Note that we calculate the sum over layers of the max number of tflops among proposers in each MoA layer as multiple proposers can run in parallel. Our plots illustrate a Pareto frontier that strikes an optimal balance between performance and cost. We show that the MoA approach lies on this Pareto front, as opposed to GPT-4 Turbo and GPT-4o which are not cost-optimal and are more expensive compared to MoA approaches of the same LC win rate. *Single Proposer*: uses the same model to generate multiple responses in each MoA layer; *Multi Proposer*: uses different models in each MoA layer. The actual tflops of GPT-4 is unknown, so we use the rumored size from the community of an 8x220B architecture.

**Cost Effectiveness**     In Figure 6a, we plot the LC win rate against the average inference cost for each instance in the AplacaEval 2.0 benchmark. The cost is calculated based on pricing information available from API provider websites.[4] This helps identify cost-effective models that achieve high performance without incurring excessive expenses. The chart reveals a Pareto front where certain models strike an optimal balance between cost and performance. Models closer to this Pareto front are more desirable as they provide better monetary value by delivering high LC win rates at lower costs. Specifically, if we prioritize the quality, MoA is the best configuration. However, if we want to strike a good balance between quality and cost, MoA-Lite can match GPT-4o's cost while achieving higher level of quality. It outperforms GPT-4 Turbo by approximately 4% while being more than twice as cost-effective. We have also benchmarked MoA using small models ($\leq$9B) as presented in Appendix A.

**Tflops Consumption**     Figure 6b depicts the relationship between LC win rate and the number of tflops. Here we use the number of tflops as a proxy for latency since latency can vary depending on the inference systems. This analysis is to understand how different models manage their budgets while maintaining or improving performance levels. Similar to the cost efficiency analysis, a Pareto front can be observed here as well. Models on this front effectively utilize their computational resources to maximize their LC win rate. For models accessed via inference endpoints, latency may not directly correlate with tflops, as other computational demands (e.g., batching) and factors (e.g., server load) can influence actual response time. However, we use tflops as an approximate indicator of the relative resource intensity of different models, as it reflects the theoretical computational requirements.

---

[4]For open-source models, we calculate the price using data from `https://api.together.ai/models`; for OpenAI models, we use pricing details from `https://openai.com/api/pricing/`. Pricing data was retrieved as of May 22, 2024.

## 4 RELATED WORK

### 4.1 LLM REASONING

In order to improve generation quality of LLMs, recent researches have experienced great progresses in optimizing LLMs to various downstream tasks through prompt engineering. Chain of Thought (CoT) (Wei et al., 2022; Kojima et al., 2022) prompting techniques represent a linear problem-solving approach where each step builds upon the previous one. Fu et al. (2022) applied CoT to multi-step reasoning tasks. To automate CoT prompting, Auto-CoT (Zhang et al., 2022b) constructs demonstrations by sampling diverse questions and generating reasoning chains. Active-Prompt (Diao et al., 2023) focuses on selecting the most uncertain questions for task-specific annotations. PS Prompt (Wang et al., 2023) decomposes tasks into subtasks. Tree-of-Thought (ToT) (Yao et al., 2023a) expands on the reasoning process by considering multiple paths of reasoning and self-evaluating choices. Effective Graph-of-Thought (Yao et al., 2023b) frames thoughts as graphs. Natural Program prompting (Ling et al., 2023) is proposed for better solving deductive reasoning tasks. And re-reading prompt (Xu et al., 2023b) revisits question information embedded within input prompts.

### 4.2 MODEL ENSEMBLE

A straightforward solution to leverage the strengths of multiple models is reranking outputs from different models. For instance, Jiang et al. (2023) introduce PAIRRANKER, which performs pairwise comparisons on candidate outputs to select the best one, showing improvements on a self-constructed instruction dataset. To address the substantial computational costs associated with multi-LLM inference, other studies have explored training a *router* that predicts the best-performing model from a fixed set of LLMs for a given input (Wang et al., 2024a; Shnitzer et al., 2024; Lu et al., 2023). Additionally, FrugalGPT (Chen et al., 2023b) proposed reducing the cost of using LLMs by employing different models in a cascading manner. In order to better leverage the responses of multiple models, Jiang et al. (2023) trained a GENFUSER, a model that was trained to generate an improved response to capitalize on the strengths of multiple candidates. Huang et al. (2024) proposed to fuse the outputs of different models by averaging their output probability distributions.

Another line of work is multi-agent collaboration. Several studies explore using multiple large language models as agents that collectively discuss and reason through given problems interactively. Du et al. (2023) establishes a mechanism for symmetric discussions among agents. Around the same time, MAD (Liang et al., 2023) introduces an asymmetric mechanism design, with different roles, i.e., debater and judge. Other similar works include (Chan et al., 2023; Xu et al., 2023c; Liu et al., 2023; He et al., 2023). Moreover, ReConcile (Chen et al., 2023a) exemplifies an asymmetric discussion involving weighted voting. To understand discussion more deeply, Zhang et al. (2023) and Chen et al. (2023c) aim to explain such collaboration mechanism in a social psychology view. Wang et al. (2024b) compared multi-agent approaches and found a single agent with a strong prompt including detailed demonstrations can achieve comparable quality to multi-agent approaches.

## 5 CONCLUSION

This paper introduces a Mixture-of-Agents approach aimed at leveraging the capabilities of multiple LLMs via successive stages for iterative collaboration. Our method harnesses the collective strengths of agents in the Mixture-of-Agents family, and can significantly improve upon the output quality of each individual model. Empirical evaluations conducted on AlpacaEval 2.0, MT-Bench, and FLASK demonstrated substantial improvements in response quality, with our approach achieving the LC win rate up to $65\%$. These findings validate our hypothesis that integrating diverse perspectives from various models can lead to superior performance compared to relying on a single model alone. In addition, we provide insights into improving the design of MoA; systematic optimization of MoA architecture is an interesting direction for future work.

**Limitations.** Our proposed method requires iterative aggregation of model responses, which means the model cannot decide the first token until the last MoA layer is reached. This potentially results in a high Time to First Token (TTFT), which can negatively impact user experience. To mitigate this issue, we can limit the number of MoA layers, as the first response aggregation has the most significant boost on generation quality. Future work could explore chunk-wise aggregation instead of aggregating entire responses at once, which can reduce TTFT while maintaining response quality.

**Broader Impact.** This study holds the potential to enhance the effectiveness of LLM-driven chat assistants, thereby making AI more accessible. Moreover, since the intermediate outputs that are expressed in natural language, MoA presented improves the interpretability of models. This enhanced interpretability facilitates better alignment with human reasoning.

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

Table 6: Results on small models. When benchmarking each aggregator, we use the following models as proposers: gemma-2-9b, llama-3.1-8b, mistral-7b-instruct-v0.3, qwen-1.5-7b.

| Aggregator | Layers | Alpaca (LC) | Alpaca (win) | Arena-Hard | MT-Bench |
|---|---|---|---|---|---|
| gemma-2-9b | 1 | 48.54 | 36.26 | 40.6 | 8.49 |
| | 2 | 56.54 | 48.20 | 47.5 | 8.44 |
| | 3 | 56.83 | 49.47 | 47.8 | 8.50 |
| llama-3.1-8b | 1 | 26.06 | 27.48 | 28.0 | 8.34 |
| | 2 | 29.52 | 38.48 | 34.9 | 8.39 |
| | 3 | 33.34 | 42.68 | 36.6 | 8.49 |
| mistral-7b-instruct-v0.3 | 1 | 19.88 | 15.67 | 16.3 | 7.59 |
| | 2 | 26.93 | 24.68 | 22.1 | 8.26 |
| | 3 | 27.98 | 27.79 | 24.1 | 8.17 |
| qwen-1.5-7b | 1 | 16.58 | 13.12 | 12.6 | 7.64 |
| | 2 | 25.54 | 24.36 | 20.7 | 7.98 |
| | 3 | 28.94 | 29.91 | 23.2 | 7.86 |

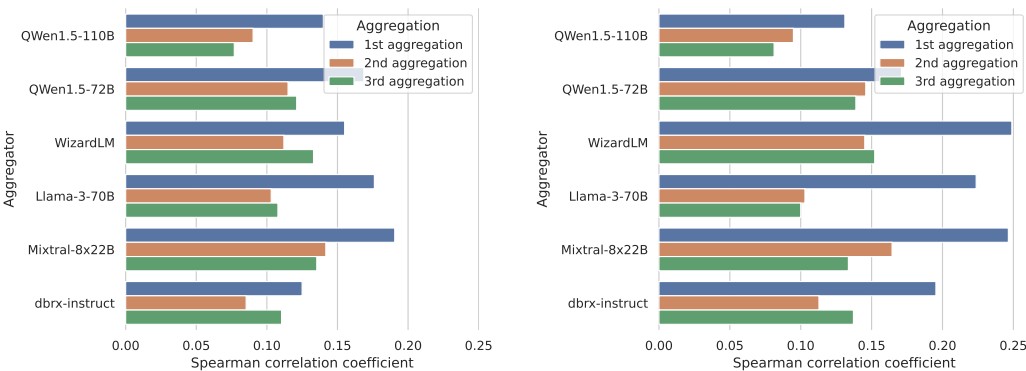

Figure 7: (a) Spearman Correlation using TF-IDF similarity; (b) Spearman Correlation using Levenshtein similarity.

## A    RESULTS ON SMALL MODELS

We have benchmarked and observed similar performance improvements with smaller models as well. As shown in Table 6, for models with ≤9B parameters, MoA can improve performance by up to 12% compared to individual models. Specifically, using gemma-2-9b as the aggregator, we achieve a 56.83% LC win rate with small models alone, outperforming GPT-4 and comparable to GPT-4o. These results demonstrate that MoA can achieve significant performance improvements even with reduced computational resources. Note that for MT-Bench sometimes 3 layer is worse than 2 layer mostly due to smaller models have less multi-turn capability. 2 layer MoA is consistently better than the original model.

## B    SPEARMAN CORRELATION USING DIFFERENT SIMILARITY FUNCTIONS

We present results using TF-IDF-based similarity and Levenshtein similarity when calculating the Spearman correlation. Specifically, within each sample of $n$ proposed answers, we calculate Spearman correlation coefficient between the $n$ similarity scores and the $n$ preference scores determined by the GPT-4-based evaluator. As shown in Figure 7, there is indeed a positive correlation between win rate and both TF-IDF similarity and Levenshtein similarity.

# C   LLM RANKER

This section introduces the setup of the LLM-Ranker used in this paper. The LLM-Ranker is designed to evaluate and rank the best output generated by some LLMs. Below presents the template for prompting the model during these evaluations.

---

**Prompt for ranking with LLMs**

You are a highly efficient assistant, who evaluates and selects the best large language model (LLMs) based on the quality of their responses to a given instruction. This process will be used to create a leaderboard reflecting the most accurate and human-preferred answers.
I require a leaderboard for various large language models. I'll provide you with prompts given to these models and their corresponding outputs. Your task is to assess these responses, and select the model that produces the best output from a human perspective.

## Instruction

```
{
    "instruction": """{instruction}""",
}
```
## Model Outputs
Here are the unordered outputs from the models. Each output is associated with a specific model, identified by a unique model identifier.

```
{
    {
        "model_identifier": "{identifier_1}",
        "output": """{output_1}"""
    },
    {
        "model_identifier": "{identifier_2}",
        "output": """{output_2}"""
    },
    {
        "model_identifier": "{identifier_3}",
        "output": """{output_3}"""
    },
    {
        "model_identifier": "{identifier_4}",
        "output": """{output_4}"""
    },
    {
        "model_identifier": "{identifier_5}",
        "output": """{output_5}"""
    },
    {
        "model_identifier": "{identifier_6}",
        "output": """{output_6}"""
    }
}
```
## Task
Evaluate the models based on the quality and relevance of their outputs, and select the model that generated the best output. Answer by providing the model identifier of the best model. We will use your output as the name of the best model, so make sure your output only contains one of the following model identifiers and nothing else (no quotes, no spaces, no new lines, ...).

## Best Model Identifier

---

Table 7: Case: Some models produce high quality answers.

| Role | Preference | Content |
|------|-----------|---------|
| User | - | I want to talk about "Smooth", a song by Rob Thomas. |
| Qwen1.5-110B-Chat | 0.35 | ... It's a perfect blend of rock, pop, and Latin music, showcasing Santana's signature guitar work and Rob Thomas's soulful vocals ... |
| Qwen1.5-72B-Chat | 0.00 | ... |
| Llama-3-70B-Instruct | 0.00 | ... it's a collaboration between Rob Thomas, the lead vocalist of Matchbox Twenty, and Santana, the legendary guitarist ... |
| WizardLM-2-8x22B | 0.99 | ... It spent 12 weeks at the top of the Billboard Hot 100 chart in the United States ... capturing the song's sultry and energetic vibe ... |
| Mixtral-8x22B-Instruct-v0.1 | 0.00 | ... |
| dbrx-instruc | 0.00 | ... |
| Aggregated (Qwen1.5-110B-Chat) | 0.99 | "Smooth" is actually a collaboration between Rob Thomas, who is the lead singer of the band Matchbox Twenty, and the renowned Latin rock guitarist Carlos Santana ... This iconic song blends rock, pop, and Latin music elements, showcasing Santana's signature guitar work and Rob Thomas's distinctive vocals ... The song was a colossal success, spending 12 weeks at the top of the Billboard Hot 100 ... The music video, set in a vibrant club scene, complements the song's sultry vibe ... |

Table 8: Case: all proposed responses are not good enough.

| Role | Preference | Content |
|------|-----------|---------|
| User | - | How do you become an author? |
| Qwen1.5-110B-Chat | 0.00 | Becoming an author involves several steps and ... 3. **Choose your niche**:... 8. **Understand the publishing industry**:... 10. **Self-publishing options**:... |
| Qwen1.5-72B-Chat | 0.00 | ... |
| Llama-3-70B-Instruct | 0.16 | Becoming an author can be a rewarding and fulfilling career... 1. **Develop a love for reading and writing**:... 10. **Market and promote your work**:... 11. **Continuously improve**:... |
| WizardLM-2-8x22B | 0.03 | 1. ... 4. Finish your work:... |
| Mixtral-8x22B-Instruct-v0.1 | 0.00 | ... |
| dbrx-instruc | 0.00 | ... |
| Aggregated (Qwen1.5-110B-Chat) | 0.33 | Becoming an author involves a dedicated journey... 1. **Cultivate a Love for Writing and Reading**:... 4. **Choose Your Niche**:... 5. **Finish Your Work**:... 10. **Self-Publishing**:... 11. **Marketing and Promotion**:... 12. **Continuous Learning and Writing**:... |

## D    Case Study

We present a case study in this section. Due to the length of the responses generated by all models, we will only show selected fragments for brevity. To illustrate how the aggregator synthesizes the response, we underlined similar expressions between the proposed responses and the aggregated response in different colors. We omit the content that all proposed responses have mentioned.

Table 7 showcases the responses generated by different proposers. The aggregated response generated by Qwen1.5-110B-Chat reflects a high preference for its own content but also incorporates key points from Llama-3-70B-Instruct and WizardLM 8x22B. Notably, GPT-4's preference score for WizardLM 8x22B's response is 0.99, and the final aggregated answer also achieves a preference score of 0.99.

Meanwhile, Table 8 presents another case where none of the proposed responses achieve a high GPT-4 preference score. Despite this, the aggregator successfully identifies and incorporates the strong points from these responses, achieving a preference score of 0.33.

## E    Reasoning Evaluations

Here, we demonstrate that our approach is applicable to reasoning tasks including the MATH dataset Hendrycks et al. (2021b), Big-Bench Hard (BBH) Suzgun et al. (2023), MMLU Hendrycks et al. (2021a) and CSQA Talmor et al. (2021). Specifically, BBH evaluates models on 23 multi-step complex reasoning tasks; MMLU contains 57 tasks on knowledge and reasoning; CSQA includes difficult commonsense reasoning questions. We posit this covers a wide range of domains including coding, math, knowledge, commonsense QA and complex reasoning.

The results are presented in Table 9, where we show that our method consistently enhances accuracy. This indicates that our approach is effective for a variety of reasoning tasks. Notably, our method is complementary to existing reasoning techniques such as Chain of Thought Wei et al. (2022) and Self-consistency Wang et al. (2022).

In addition, we investigate whether more layers in MoA can further increase performance on the MATH dataset. As shown in Table 10, MoA with three layers improves from having just two layers.

Table 9: Performance comparison across reasoning benchmarks.

| Model | BBH | MMLU | CSQA | MATH | Average |
|---|---|---|---|---|---|
| Qwen1.5-72B-Chat | 0.619 | 0.6931 | 0.8231 | 0.428 | 0.641 |
| Qwen1.5-110B-Chat | 0.6733 | 0.7624 | 0.8346 | 0.500 | 0.693 |
| Wizard 8x22b | 0.7461 | 0.7989 | 0.7871 | 0.544 | 0.719 |
| Mixtral-8x22B-Instruct-v0.1 | 0.6693 | 0.7821 | 0.8075 | 0.282 | 0.635 |
| Llama-3-70b-chat-hf | 0.7438 | 0.7978 | 0.8305 | 0.456 | 0.707 |
| dbrx-instruct | 0.3552 | 0.6867 | 0.7625 | 0.314 | 0.530 |
| MoA-Lite | **0.7667** | **0.8268** | **0.8444** | **0.570** | **0.752** |

Table 10: Results on the MATH task. We evaluate different aggregators, with all six models serving as proposers in each MoA layer.

| Aggregator | Layer 1 | Layer 2 | Layer 3 |
|---|---|---|---|
| Qwen1.5-72B-Chat | 0.428 | 0.526 | 0.552 |
| Qwen1.5-110B-Chat | 0.500 | 0.570 | 0.576 |
| Wizard 8x22b | 0.544 | 0.574 | 0.580 |
| Mixtral-8x22B-Instruct-v0.1 | 0.282 | 0.534 | 0.556 |
| Llama-3-70B-Instruct | 0.456 | 0.584 | 0.578 |
| dbrx-instruct | 0.314 | 0.456 | 0.522 |

Table 11: Performance comparison of MoA-Lite and MoA searched using our proposed optimization method.

| Model | Aggregate | AlpacaEval (LC) | Arena-Hard | MT-Bench |
|---|---|---|---|---|
| MoA-Lite | 74.1 | 59.3 | 71.3 | **9.18** |
| MoA-Lite searched | **75.0** | **62.0** | **71.8** | 9.11 |

## F  SEARCH FOR AN OPTIMAL MoA ARCHITECTURE

We found an automatic architecture optimization method convenient for practical use, especially when new models are introduced. We implemented a relatively basic optimization method to select the set of LLMs as a proof of concept.

**Setup**  Specifically, we fix the number of layers to be two and the aggregator to be Qwen-1.5-110b-Chat, and set the number of models and which model in proposers to be variables for optimization. We utilized Broyden–Fletcher–Goldfarb–Shanno algorithm (BFGS) for this unconstrained optimization problem.

**Validation Data**  It is important to have a good set of validation data. We randomly sampled 50 problems from AlpacaEval and 50 from Arena-Hard. The combined size of 100 enables us to verify architecture performances quickly. We averaged the scores of AlpacaEval and ArenaHard to be our final metric.

We ran the optimization and found the best mixture to be WizardLM-2-8x22b, Qwen-1.5-110b-Chat, Qwen-1.5-72b-Chat, and three Llama-3-70b-Instruct as proposers and Qwen-1.5-110b-Chat as aggregator. The resulting mixture outperforms our MoA-Lite on two out of the three benchmarks as shown in Table 11.

## G  INDIVIDUAL PROPOSER'S SIMILARITY TO AGGREGATOR

We conducted an ablation study to determine whether the aggregator merely repeats the outputs of the proposers or performs additional processing. To investigate this, we calculated the similarity between the outputs of individual proposers and the aggregator using BLEU scores (considering 3-grams, 4-grams, and 5-grams).

The results, shown in Table 12, indicate that the aggregator tends to exhibit the highest similarity with its own generation (Qwen1.5-110B-Chat). However, the overall similarity scores remain relatively low, suggesting that the aggregator is not simply replicating the proposers' outputs but is instead synthesizing or refining them.

Additionally, we observed that as the layer depth increases, the similarity between the aggregator and the proposers also increases. This indicates that responses across layers grow more aligned, potentially reflecting convergence toward a consensus or refinement as the sequence progresses.

## H  GPT MODELS USED

Here's a consolidated list of the GPT-family models used across the experiments:

- **GPT-3.5-turbo-0125**: Referenced in Figure 3.

- **GPT-4 Preview**: Referenced in Figure 5.

- **GPT-4-turbo**: A more cost-effective variant of GPT-4 with improved latency and efficiency, referenced in Figure 6a and Figure 6b.

- **GPT-4o**: We use gpt-4o-2024-0513 across the paper.

Table 12: Similarity scores between proposers and aggregators in MoA-Lite and MoA settings.

| Aggregator | Score |
|---|---|
| *MoA-Lite (Qwen1.5-110B-Chat)* | |
| Qwen1.5-110B-Chat | 0.3972 |
| WizardLM-2-8x22B | 0.3742 |
| Qwen1.5-72B-Chat | 0.3528 |
| Llama-3-70B-Instruct | 0.3230 |
| Mixtral-8x22B-Instruct-v0.1 | 0.3149 |
| dbrx-Instruct | 0.3062 |
| *MoA (Qwen1.5-110B-Chat)* | |
| Qwen1.5-110B-Chat | 0.5102 |
| WizardLM-2-8x22B | 0.4539 |
| Qwen1.5-72B-Chat | 0.4009 |
| Llama-3-70B-Instruct | 0.3827 |
| Mixtral-8x22B-Instruct-v0.1 | 0.3730 |
| dbrx-Instruct | 0.3640 |

# I    EFFECT OF PROPRIETARY MODELS

In this section, we investigate whether a bit more in-depth into the incorporation of gpt-4o-2024-05-13 in MoA. Specifically, we try to incorporate it both as an aggregator and as a proposer. When using GPT4-o as a proposer, replacing dbrx-Instruct, we can see a clear boost in performance for AlpacaEval and Arena-Hard scores while maintaining MT-Bench score demonstrated in Table 13. Using GPT-4o as an aggregator increases the performance significantly. The aggregate score jumps from 74.1 to 83.3.

Table 13: When incorporating GPT-4o into MoA, we can see clear benefits in terms of benchmark improvements. Although more improvements come from using GPT-4o as the aggregator. We use gpt-4o-2024-05-13 for this experiment.

| Model | Agg. | AlpacaEval 2.0 (LC) | Arena-Hard | MT-Bench |
|---|---|---|---|---|
| MoA-Lite | 74.1 | 59.3 | 71.3 | 9.18 |
| MoA-Lite w/ GPT-4o replacing dbrx | 75.7 | 63.1 | 73.1 | 9.1 |
| MoA-Lite w/ GPT-4o | **83.3** | **65.7** | **90.3** | **9.4** |

