# OpenReview forum: "Mixture-of-Agents Enhances Large Language Model Capabilities"
_ICLR.cc/2025/Conference — ICLR 2025 Spotlight_

### Official Review · Reviewer_or3e · 2024-10-25

**Soundness:** 3
**Presentation:** 4
**Contribution:** 3
**Rating:** 8
**Confidence:** 3

**Summary:**

The paper proposes a method to use multiple LLMs as agents to improve the quality of generated responses to arbitrary queries. Agents are deployed in mixture-of-agent (MoA) layers. In the first MoA layer, multiple agents generate an answer to the input query independently. In subsequent layers, each agent in layer $T$ generates an answer to the query while also seeing the answers of all agents from layer $T-1$. At the last layer, a single aggregator LLM is used to combine all answers generated by the agents in the last MoA layer into a single answer. The proposed method shows state-of-the-art performance on multiple benchmarks compared to recent models from the GPT family including GPT4-omni, GPT4-turbo.

**Strengths:**

- The idea of using multiple LLMs as agents to generate answers and to aggregate these answers in a sequence of mixture-of-agents (MoA) layers is interesting.
- Very good performance compared to many relevant baselines.
- Experiments are solid and well conducted, and the ablation studies give important insight into the most relevant aspects of the proposed method.

**Weaknesses:**

- The main weakness, in my opinion, is the fact the authors did not use human evaluation at any parts of their pipeline. I understand that LLMs can perform very well (according to measures of correlation to humans) on multiple benchmarks, but at least one small-scale human evaluation could have been provided as a sanity-check.

**Questions:**

- Do you use humans in your evaluation in any part of your paper? It seems that all your tests use (language) models for evaluation. Did you conduct any (manual/human) analysis of the outputs generated by the models at any stage?
- An experiment that could be more prominently contextualised is the case where there is one aggregator/proposer agent in each layer (numbers in Table 3, discussion in **Effect of model diversity and the number of proposers**. This is kind of a middle-case between the LLM-as-reranker baseline (which you clearly outperform) and your proposed idea in full generality. In the case of a single proposer, your win rate in Table 3 is 47.8% (so not better than the baseline). I think it would be very valuable to include this setting as a baseline also in all other experiments and not only those for AlpacaEval2.0, since this really shows when the proposed method starts to show improvements to the baselines.
- The ablations are very informative and well conducted, and covers all the important points one would want to see in the main paper.
- Across the paper, there are mentions to GPT4 Preview (Figure 5a), GPT-3.5-turbo-0125 (Figure 3), GPT4-turbo (Figure 6a), GPT4-omni (in multiple places). Could you please introduce all the models from the GPT family you used across your experiments in a single place with an unambiguous description of their model name if available? The paper is in general very well organised, this is however one of the few points where this organisation could be improved.
- In Figures 6a and 6b, please include in your description why you have multiple circles of the same size/colour, and provide some context on what you mean by '1', '2', '3'. For someone who read the paper and reach the figures that is very clear, but someone looking at the figure without that context fresh may not understand it. Also, is it correct to assume that each of the large blue circles (MoA, multi-proposer, with 3 MoA layers) is an MoA model with different configurations (like different LLMs as final aggregators)? If that is the case, it may make sense to add some nuance to your discussion of the results. In practice, for someone with a new problem who want to use your paper's idea, do you assume by reusing your best-performing MoA, they will also get the best results on their new problem? In other words: should you not include also something like the average over the 6 large blue circles (or perhaps over the 3 best large blue circles)?

---

### Official Review · Reviewer_YS2v · 2024-11-04

**Soundness:** 4
**Presentation:** 3
**Contribution:** 3
**Rating:** 8
**Confidence:** 4

**Summary:**

This work presents a Mixture of Agents (MoA) Architecture. This multi-layer architecture harnesses the collaborativeness of LLMs, where agents (models) in later layers aggregate and combine auxiliary information from previous layers with the user task to yield the final output. They evaluate this proposed architecture on AlpacaEval 2.0, Arena-Hard, MT-bench, and Flask and show that their architecture performance surpasses GPT-4o.

The "collaborativeness" of LLMs is defined as the phenomenon where the LLM generates better responses when presented with outputs from other models, even if these models are less capable.

The architecture presented has two types of models: "proposer models," which generate useful references for use by other models; these models offer "perspective" or might even act as auxiliary/additional information providers to the aggregator. The other type of models present are "aggregator models," which are proficient in combining responses from other models into a single high-quality output.

They not only show an MoA architecture but also a MoA-Lite architecture that outperforms GPT-4o and is both better and more cost-effective. They conduct various empirical experiments showcasing the effectiveness of their approach to MoA outperforming LLM-Ranking, the ability to incorporate best answers, and the effects of model diversity and different proposers.

**Strengths:**

- While collaborativeness has been harnessed in various ways, a layered funnel architecture in which earlier layers add information for later layers to consume and interplay to summarize these outputs efficiently to yield a final output has not been explored.

- The authors also thoroughly conducted their experiments to establish collaborativeness and benchmark various datasets. The usage of open-source models to showcase the results helps to make replicating these results possible.

- They also analyze the cost-effectiveness of these models compared to some base models, showing the effectiveness of applying this architecture to obtain better results at a cheaper rate.

**Weaknesses:**

- The idea of collaborativeness or hierarchical processing in LLMs is not exactly novel [1][2]; if you think of different layers in the architecture using the same model, this reduces to some form of iterative refinement of outputs as shown in [2].

- Some of the analysis in the paper to support budget analysis is unclear.



#### *References*
1] [2308.10848] AgentVerse: Facilitating Multi-Agent Collaboration and Exploring Emergent Behaviors

2] LEGO: A Multi-agent Collaborative Framework with Role-playing and Iterative Feedback for Causality Explanation Generation - ACL Anthology

**Questions:**

1] What is the point of the tflops consumption analysis? And where are the numbers for these obtained from? Gpt-4o and Gpt-4-turbo are closed-source models, so it is unclear where those numbers are obtained.

2] In a similar vein, can tflops really be a proxy for latency where one is used to perhaps look at real-time serving of models while the other looks at consumption? Especially when the system in question is not locally run but uses an inference endpoint (as mentioned in section 3.1)

3] While you analyzed a model's propensity to be an aggregator or a proposer, did you consider combining models that might be good at different kinds of tasks (for example, math or science)?

4] Was there any reason to include LLM reasoning in the related work when the work does not explicitly show gains on any reasoning dataset and the evaluation is mostly on instruction following/preference benchmarks?

Last but not least, just a suggestion for the paper's readability: it would significantly enhance the analogy to MoE if there was an explicit mention that MoE uses routing/gating networks to sometimes choose between routing between single or multiple experts. They specialize in different aspects vs traditional networks where all input data is processed through every layer. There is a mention in the work that the MoA, from the specification of the architecture, consolidates the role of the gating network and expert network using a LLM. Still, it is not the case on some level because it is not choosing which experts it will route its output from, and the input does pass through all layers, the aggregator picks and chooses information from the proposer LLMs but it does not necessarily always just drop some inputs.

---

### Official Review · Reviewer_NT9P · 2024-11-05

**Soundness:** 3
**Presentation:** 3
**Contribution:** 3
**Rating:** 8
**Confidence:** 4

**Summary:**

The paper proposes Mixture-of-Agents (MoA) as a possible pathway to leverage the collective strengths of multiple LLMs.  By stacking LLMs into multiple layers and employing the output of LLM in previous layers, the frameworks allows a iterative refinement and strike a balance between inference cost and the performance.

**Strengths:**

+ Proposal of a new effective framework to employ the collective intelligence of multiple LLMs.
+ Empirical evaluation on AlpacaEval 2.0, Arena-Hard, and MT-Bench verifies the effectiveness of the proposed solution.

**Weaknesses:**

+ Stacking LLMs into layers and revising the output obtained from previous layers seems like another form of model ensemble and I would suggest including model ensemble as one of the comparative methods.
+ In Figure 6, the max number of tflops among proposers in each MoA layer is used as an approximation of the total tflops of the entire layers since different proposers can run in a parallel way. However, the approximation is only reasonable when considering the inference latency for a single query. Otherwise, if we have a batch of data, with the same computation resources MoA can introduce extra latency since the parallel of proposers limits the data parallel size.
+ The evaluation of MoA is only performed on instruction-following benchmarks. More evaluation of reasoning and commonsense knowledge benchmarks such as MMLU and BBH is necessary for a comprehensive understanding of the efficacy of MoA.
+ Detailed ability on the specific role of each proposer is missing. How does each proposer contribute to the final output? Is it possible for a proposer to simply repeat the output from previous layers? Will the models in later layers revise the mistakes made in previous layers?

**Questions:**

See the weaknesses above.

---

### Official Review · Reviewer_U7wn · 2024-11-11

**Soundness:** 2
**Presentation:** 2
**Contribution:** 2
**Rating:** 6
**Confidence:** 4

**Summary:**

This paper proposes a mixture-of-agents (MoA) framework that leverages the collective insights of multiple large language models (LLMs).

The core process involves several iterative steps: (1) multiple LLMs generate initial draft proposals; (2) these proposals are shared with all LLMs in the following iterations to generate refined proposals; (3) this second step is repeated multiple times, with an aggregator ultimately producing the final response.

Experimental results on four benchmarks (AlpacaEval 2.0, MT-Bench, Arean Hard, and FLASK) demonstrate that MoA can outperform GPT-4 Omni.

**Strengths:**

1. To the best of my knowledge, the proposed framework is both novel and reasonable. MoA can be viewed as a specific method for combining multiple weaker models to create a stronger model.

2. The model's performance is competitive, showing improvements over GPT-4 Omni on three benchmarks.

**Weaknesses:**

1.  The proposed model is more resource-intensive than single LLM-based models.
2.  Most evaluations use LC metrics, with only a limited evaluation on MATH tasks included in the appendix. Further evaluations on diverse tasks are necessary to illustrate the general advantages of the proposed method.

3. An important question is how to select the set of proposal LLMs. Currently, the paper demonstrates two setups: one with relatively large models and one with smaller models. However, there is no guideline or formal strategy for choosing LLM sets. Additionally, it would be interesting to see the performance of a mixed set containing a small number of large LLMs and a majority of small LLMs.  Combining models with different language data is also a very interesting topic. Another crucial question is determining the number of layers. Aside from treating it as a hyperparameter, is there a way to determine an optimal layer count for a given set of proposal LLMs?

4. GPT-4o is only used as the aggregator. It would be informative to evaluate its performance if it were included as one of the proposal LLMs.

**Questions:**

1.  Why do you call your method as mixture of agents instead of mixture of LLMs.   I did not see any behaviours of agents.

2.  Are there any lines missing in Figure 6 (a) and (b)?

---

### Meta-Review · Area_Chair_EcGM · 2024-12-20

**Metareview:**

The paper introduces a Mixture-of-Agents (MoA) approach that combines multiple large language models (LLMs) to improve performance. The MoA architecture is layered, with each layer containing multiple LLM agents. Each agent utilizes outputs from the previous layer's agents as auxiliary information to generate its response. The paper demonstrates comprehensive and strong results of MoA against many strong LLMs on various benchmarks. There are some comments from the reviewers, e.g. the searching efforts to select proposal LLMs, unclear budget analysis, and human evaluation results. I encourage the authors to incorporate those comments into the final paper.

**Additional Comments On Reviewer Discussion:**

n/a

---

### Decision · Program_Chairs · 2025-01-22

Accept (Spotlight)